# Biodegradation of Poly(ethylene terephthalate) by *Bacillus safensis* YX8

**DOI:** 10.3390/ijms242216434

**Published:** 2023-11-17

**Authors:** Caiting Zeng, Fanghui Ding, Jie Zhou, Weiliang Dong, Zhongli Cui, Xin Yan

**Affiliations:** 1College of Life Sciences, Nanjing Agricultural University, Nanjing 210095, China; 2020116069@stu.njau.edu.cn (C.Z.); 2022116053@stu.njau.edu.cn (F.D.); czl@njau.edu.cn (Z.C.); 2College of Biotechnology and Pharmaceutical Engineering, Nanjing Tech University, Nanjing 211816, China; jayzhou@njtech.edu.cn (J.Z.); dwl@njtech.edu.cn (W.D.); 3Jiangsu Provincial Key Laboratory for Organic Solid Waste Utilization, Nanjing Agricultural University, Nanjing 210095, China

**Keywords:** poly(ethylene terephthalate), PET waste, biodegradation, *Bacillus safensis*

## Abstract

Due to the extensive utilization of poly (ethylene terephthalate) (PET), a significant amount of PET waste has been discharged into the environment, endangering both human health and the ecology. As an eco-friendly approach to PET waste treatment, biodegradation is dependent on efficient strains and enzymes. In this study, a screening method was first established using polycaprolactone (PCL) and PET nanoparticles as substrates. A PET-degrading strain YX8 was isolated from the surface of PET waste. Based on the phylogenetic analysis of 16S rRNA and *gyrA* genes, this strain was identified as *Bacillus safensis*. Strain YX8 demonstrated the capability to degrade PET nanoparticles, resulting in the production of terephthalic acid (TPA), mono (2-hydroxyethyl) terephthalic acid (MHET), and bis (2-hydroxyethyl) terephthalic acid (BHET). Erosion spots on the PET film were observed after incubation with strain YX8. Furthermore, the extracellular enzymes produced by strain YX8 exhibited the ability to form a clear zone on the PCL plate and to hydrolyze PET nanoparticles to generate TPA, MHET, and BHET. This work developed a method for the isolation of PET-degrading microorganisms and provides new strain resources for PET degradation and for the mining of functional enzymes.

## 1. Introduction

PET is one of the most commonly used polyester plastics in the world because of its non-toxic, odorless, high mechanical strength and stability [1,2]. Although PET has brought great convenience to us, the accumulation of PET waste in the environment has polluted the sea and soil, posing a serious threat to aquatic life and even human health [3,4]. Additionally, PET waste could break down into microplastics with a diameter of less than 5 mm, which can spread through the food chain to humans [5,6]. Currently, PET waste is treated by landfill, incineration, mechanical treatment, chemical, and biodegradation [7]. Among these methods, biodegradation can depolymerize PET to yield compounds, including BHET, MHET, TPA, and ethylene, which can be used to re-synthesize PET or produce high-value-added products [8,9,10]. As a green and low-cost method for PET waste disposal, biodegradation has irreplaceable advantages and broad development prospects [11,12].

Following years of work, a number of PET-degrading strains were isolated, and PET-degrading enzymes were purified from them [13,14,15,16,17,18]. Thermobifida fusca hydrolase (TfH) was purified from Thermobifida fusca, which could break down 50% of PET bottles with 10% crystallinity within 3 weeks at 55 °C [19]. A cutinase called HiC was extracted from *Humilica insolens*, which caused 97% weight loss of low crystallinity PET films (7%) at 70 °C for 6 days [20]. With advancements in technology, metagenomics was applied to the mining of PET-degrading enzymes. The LC-cutinase (LCC) was cloned and expressed with high thermal stability (half-lives of 40 min at 70 °C) and catalytic activity (12 mg/h/mg of PET-degrading activity) from leaf-branch compost using a metagenomic approach [21]. It is worth noting that the isolation of *Ideonella sakaiensis* 201-F6 from 250 samples could use PET with 1.9% crystallinity as the major carbon source for growth. The PET-degrading enzyme PETase was also extracted from it, which was able to completely degrade PET film with 1.9% crystallinity within 6 weeks at 30 °C and produce BHET, MHET, and TPA [22]. This is the first enzyme with high PET degradation ability at low temperatures.

Although a variety of PET-degradation strains and enzymes have been explored, the high degree of polymerization and resistance to biodegradation of PET has made it challenging to develop efficient and effective biological methods for PET recycling [23,24,25,26]. It is still necessary to explore more novel PET-degrading strains and enzymes to enrich PET-degrading resources [27]. The enzymes and strains that exhibit outstanding ability to degrade crystallized PET or to work efficiently at moderate temperatures are desired. In this study, a PET-degrading strain YX8 was isolated from the surface of PET waste. Based on phylogenetic analysis, the phylogenetic status of the strain YX8 was identified, and its degradation characteristics of PET nanoparticles and PET film were investigated using degradation product detection, scanning electron microscopy (SEM) observation, and water contact angle (WCA) detection. In addition, the PET degradation ability of extracellular enzymes was tested.

## 2. Results

### 2.1. Isolation of PET Degrading Bacteria

The low chain mobility, high polymerization, and high hydrophobicity of PET limit its degradation by microorganisms, which also makes the screening of PET-degrading strains or enzymes labor-intensive and time-consuming [28]. Therefore, it is necessary to develop appropriate screening methods to accelerate the isolation of degrading strains or enzymes [29]. In this study, PCL, a polyester compound with a similar structure to PET, was used as a substrate for the preliminary screening (Figure 1). The microorganisms associated with PET waste were spread on the LB plate containing PCL. If a strain can hydrolyze PCL, a transparent halo will be formed around its colony [30]. These PCL-degrading strains were then subjected to a PET-degrading ability assay. This assay used PET nanoparticles as the substrate, and the degradation products were analyzed using high-performance liquid chromatography (HPLC).

The HPLC result showed that a strain named YX8 could convert PET nanoparticles to three products: BHET, MHET, and TPA (Figure 2). The retention time of product I (9.300 min) was identical to that of the TPA standard (9.300 min); the retention time of product II (10.240 min) and product III (10.763 min) was close to that of MHET (10.233 min) and BHET (10.690 min), respectively. According to the standard curves of corresponding standards, the concentration of products I, II, and III were 4.9, 3.3, and 0.4 μM, respectively. It was difficult to recover PET nanoparticles, so the weight loss of PET nanoparticles was not detected here.

These PET nanoparticle degradation products were further analyzed using LC-MS/MS. As shown in (Figure 3), the molecular ion mass of metabolite A was *m*/*z* 165.0190 [M-H]^+^, which was in good agreement with that of TPA (*m*/*z* 165.0193 [M-H]^+^) with a −1.8 ppm error. The molecular ion mass of metabolite B was *m*/*z* 209.0451 [M-H]^+^, which was in line with that of MHET (*m*/*z* 209.0456 [M-H]^+^) with a −2.4 ppm error. Generally, a mass error between +5 ppm and −5 ppm is acceptable [31], so metabolite A and metabolite B were identified as TPA and MHET, respectively. BHET was not detected using LC-MS/MS, probably due to its low concentration in the sample. In addition, when strain YX8 and PET nanoparticles were incubated in the MSM medium, no product was detected using HPLC, indicating that strain YX8 was unable to use PET or its intermediates for growth.

### 2.2. Identification of Strain YX8

The colonies of strain YX8 on the LB plate were off-white, opaque, round, neat, and moist (Figure 4a). It was a Gram-positive (Figure 4b), sporulating and flagellating bacterium (Figure 4c). The 16S rRNA gene sequences alignment analysis (Figure 4d) revealed that strain YX8 had more than 99.5% similarity to *Bacillus* spp., such as *B. safensis*, *B. pumilus*, and *B. altitudinis*, with the highest similarity to the type strain *B. safensis* FO-36b (100%). However, due to the close kinship of *Bacillus* spp., it is difficult to distinguish them accurately via the 16S rRNA gene only. It was found that the high variability of *gyrA* sequences gives it a stronger phylogenetic resolution for *Bacillus* identification [32]. *gyrA* sequence analysis (Figure 4e) showed that strain YX8 had the highest similarity with *B. safensis* FO-36b (96.34%). Combined with the observation of colony morphology, strain YX8 was finally identified as a member of *B. safensis*. The sequences of the 16S rRNA gene and *gyrA* gene are provided in the Appendix A.

### 2.3. Biodegradation of PET Film by Strain YX8

Commercial PET usually has crystallinity, which is more difficult to degrade than PET nanoparticles. Therefore, the degradation ability of strain YX8 against commercial PET film was tested to detect whether YX8 could degrade PET film with high crystallinity. After incubated with strain YX8 for 2 months, the surface of the PET film turned white and exhibited obvious erosion marks (Figure 5a). The result of SEM showed that there were multiple erosion spots on the surface of the film, and there were multiple massive structures on the surface of the erosion spots, which were presumed to be caused by PET depolymerization. At 5000 times magnification, it was observed that the erosion spot was clearly demarcated from the surrounding area, and further magnification revealed that the PET film was cracked at these erosion spots (Figure 5b). The WCA of PET film surface increased from 69.77° to 79.30° after incubation with strain YX8 (Figure 5c). However, no degradation products such as TPA, MHET, and BHET were detected using HPLC. Additionally, the weight loss rate of PET films in the treatment with strain YX8 was slightly higher than that in the control but not significant (Appendix A). These results showed that strain YX8 exhibited very low degradation ability toward PET film.

### 2.4. Biodegradation of PET Nanoparticles via the Extracellular Enzymes of Strain YX8

To confirm that the degradation of PCL or PET nanoparticles was catalyzed using the YX8 secrete PET degradation enzymes, the extracellular crude enzymes of strain YX8 were used to degrade PCL and PET nanoparticles. Esterase enzyme activity assay indicated that extracellular esterase activity was highest at 72 h (Appendix A), so the extracellular enzymes at this time were concentrated and incubated with PCL plate and PET nanoparticles. The concentrated enzymes exhibited the ability to form a clear zone on the PCL plate within 12 h (Appendix A).

The HPLC analysis showed that the degradation products TPA (2.9 × 10^−3^ μM), MHET (0.1 μM), and BHET (22.3 × 10^−3^ μM) were detected after incubation of the concentrated enzymes with PET nanoparticles (Figure 6). In addition, the boiled enzymes failed to degrade PCL or PET nanoparticles. These results demonstrated that the degradation of PET nanoparticles via strain YX8 was catalyzed by some extracellular enzymes.

## 3. Discussion

In this work, we first established a strategy to screen PET-degrading strains. Then, a bacterium *B. safensis* YX8, was isolated from the soil on the surface of PET waste. Strain YX8 could depolymerize PET nanoparticles to generate MHET, BHET, and TPA but failed to grow on PET. Strain YX8 exhibited low activity toward crystalline PET film. The enzymes responsible for the depolymerization of PET were secreted into the medium. This is the first strain from *B. safensis* that can degrade PET.

Finding new classes of PET-degrading strains and enzymes is important to the biodegradation and industrial recycling of PET waste [27]. The main way to discover PET-degrading enzymes is by mining them from GenBank through sequence blasts or machine learning [24,33]. This method is efficient but has a weak possibility of finding new PET-degrading enzymes. Another method to obtain PET-degrading enzymes is based on strain isolation. PET-degrading strains are first isolated, and then the enzymes are identified from these strains. This method is time- and labor-consuming but with a strong possibility to obtain new enzymes. However, the isolation of PET-degrading strains now suffers from the lack of a high throughput method. Since PET waste is very resistant to biodegradation and rare PET-degrading strains can utilize this polymer as a sole carbon source for growth, it is very difficult to enrich the PET-degrading strains using PET as a sole source of carbon and energy. So far, only one PET-degrading strain (*I. sakaiensis* 201-F6) could grow on PET, which was isolated using PET as the sole source of carbon and energy [22]. Another method to obtain PET-degrading strains is to screen candidate strains using oligomeric PET as a substrate. LB plates containing insoluble oligomer [bis(benzoyloxyethyl) terephthalate (3PET)] were employed by some researchers [34]. Suppose a strain was able to hydrolyze 3PET; a transparent halo formed around its colony. Some 3PET-hydrolyzing strains had the ability to depolymerize PET. This method is more efficient than the enrichment strategy, but 3PET is an oligomer and not commercially available. Previous work shows that some PCL-hydrolyzing enzymes could degrade PET [24,33]. Moreover, PCL is a polymer and easily purchased. Therefore, this work used PCL as a substrate for preliminary screening. The ability of the strains to degrade PET was verified using PET nanoparticles and HPLC analysis. PET nanoparticles have low crystallinity and high specific surface area, which are more readily biodegradable than PET film [35]. This method is sensitive to PET-degrading strains, which will benefit the exploration of PET-degrading strains and enzymes.

Before this work, diverse PET-degrading strains and enzymes are described and listed in Appendix A. A minority of these strains and enzymes were obtained via strain isolation. Most of them were found via sequence analysis. Generally, these strains and enzymes showed low activity against PET film. Some engineered enzymes, such as LCC^ICCG^ and FAST-PETase, exhibited outstanding ability to depolymerize PET film [36,37]. Compared to previous strains, the degrading ability of *B. safensis* YX8 toward PET film was very low. It only caused obvious erosion traces on the surface of the PET film, and no significant weight loss was found. One possibility is that the enzymes responsible for PET depolymerization in strain YX8 have low activity toward crystallized PET. Another reason is that the expression level of these enzymes is very low. The potential of these enzymes in PET recycling will be evaluated after their purification. Additionally, the WCA of PET film was increased after treatment via strain YX8, contrary to previous reports [38,39]. It is speculated that the increase in roughness on the surface led to the rise in hydrophobicity [40].

## 4. Materials and Methods

### 4.1. Chemicals and Media

PCL, TPA, and BHET were purchased from Macklin Biochemical Co., Ltd. (Shanghai, China); MHET and trifluoroacetic acid were purchased from Aladdin Biochemical Technology Co., Ltd. (Shanghai, China). PET film (product code: ES301450, 16% crystallinity) was purchased from Goodfellow (Huntingdon, UK) [41]. Lysogeny broth (LB) [42] was used to culture bacteria, while a mineral salt medium (MSM) was used to wash out and dilute the sample [43]. The LB medium contains tryptone 10 g/L, yeast extract 5 g/L, and NaCl 5 g/L. The MSM medium contains NH_4_NO_3_ 1.0 g/L, KH_2_PO_4_ 0.5 g/L, K_2_HPO_4_ 1.5 g/L, NaCl 0.5 g/L, and MgSO_4_·7H_2_O 0.2 g/L.

### 4.2. Preparation of PET Nanoparticles

PET nanoparticles were prepared according to the method described by the previous report [44]. Briefly, the bottom of the PET bottle was cut into pieces, and 1 g of PET pieces less than 0.2 mm were dissolved in 10 mL of 90% trifluoroacetic acid, stirred at 50 °C until completely dissolved, and then left overnight. The next day, 10 mL of 20% trifluoroacetic acid was added and stirred for 2 h, followed by overnight incubation. After centrifugation at 10,000× *g* for 1 h, the supernatant was discarded. Subsequently, 100 mL of 0.5% sodium dodecyl sulfate (SDS) was added and stirred vigorously. The upper layer of a 50 mL suspension was taken after leaving for 1 h, which were PET nanoparticles.

### 4.3. Isolation of PET Degrading Bacteria

PET waste was collected from a landfill plant in Xuancheng City, Anhui Province, China. The soil on the surface of PET waste was washed out and diluted with MSM and then spread on LB plates containing 0.44% PCL (PCL was dissolved in dimethyl sulfoxide at a concentration of 2.2%). The plates were then incubated at 45 °C to screen the strain that was able to form a transparent zone around its colony. The strain that showed a transparent zone was inoculated into a 50 mL LB broth containing 15 mg PET nanoparticles and grown at 45 °C, 200 rpm for 15 days. PET nanoparticles incubated without strain were used as a control. Three replicates were set up for each treatment.

After 15 days of culture, the cultures were centrifuged. The supernatant of the cultures was adjusted to pH 2.0 by adding HCl, and then twice the volume of ethyl acetate was added. Dry the upper solution in the fume hood. The sample was resuspended with methanol, filtered with a 0.22 mm membrane, and then analyzed using HPLC (Thermo Dionex UltiMate 3000, Thermo Fisher Scientific, Waltham, MA, USA) and a liquid chromatography-tandem mass spectrometer (LC-MS/MS). The instrument used in LC-MS equipped the above HPLC instrument with an electrospray ionization probe connected to a Thermo LTQ Orbitrap XL hybrid mass spectrometer (Thermo Fisher Scientific, Waltham, MA, USA). LC-MS assay conditions were the same as in HPLC, and sample data analysis was performed in negative ion mode.

All analyses were operated at room temperature (25 °C). An Eclipse Plus-C18 column (3 µm, 4.6 mm × 250 mm) was used. For the mobile phase, buffer A (0.1% formic acid in distilled water) and buffer B (acetonitrile) were used at a flow rate of 0.8 mL/min. The mobile phase was changed gradually from 85% buffer A to 50% buffer A at 12.5 min (all in vol%). The chemicals (BHET, MHET, and TPA) were detected at 240 nm [45]. The strain that could convert PET nanoparticles to BHET, MHET, or TPA was chosen for further study. Three replicates were set for each set of treatments.

### 4.4. Identification of Strain YX8

The morphology of strain YX8 was observed using a light microscope (Gram staining) and transmission electron microscopy (TEM). To prepare the TEM sample, strain YX8 was streaked on an LB plate and incubated overnight at 45 °C; the cells were suspended in ultrapure water and dripped onto a carrier net that contained a supporting membrane; the sample was stained with phosphotungstic acid (PTA) for 1 min; the cell morphology was finally observed using TEM (Hitachi HT7800, Hitachi, Tokyo, Japan). The total genomic DNA of strain YX8 was extracted according to the method described by the previous report [46]. The 16S rRNA gene was amplified via PCR (LongGene T20 multi-block thermal cycler) using the universal primers 27F (5′-AGAGTTTGATCCTGGCTCAG-3′) and 1492R (5′-TACGGTTACCTTGTTACGACTT-3′) [47]. The DNA fragment of *gyrA* was amplified via PCR using the primers gyrA-F (5′-GCDGCHGCNATGCGTTAYAC-3′) and gyrA-R (5′-ACAAGMTCWGCKATTTTTTC-3′) [32]. The PCR procedure for the above two reactions was pre-denaturation 95 °C 5 min (one round of cycle); denaturation 95 °C 15 s, annealing 56 °C 30 s, extension 72 °C 30 s/kb (29 rounds of cycles); and final extension 72 °C 10 min (1 round of cycle). DNA synthesis and sequencing were performed via Songon Biotech (Shanghai, China). The phylogenetic trees were constructed with the neighbor-joining method using MEGA6 and generated with 1000 replicates; multiple alignment analysis was performed with ClustalW [48].

### 4.5. Biodegradation of PET Film

To keep sterile, PET films were washed with 2% SDS, soaked in 70% ethanol for 4 h, and washed three times with sterile water. Strain YX8 was incubated in a 50 mL LB broth with one piece of (2 cm × 2 cm) PET film at 45 °C and 200 rpm for 2 months. The PET film incubated without strain was used as a control. Three repeats were set up. Fresh LB broth was added weekly to keep the constant volume. The products were detected using HPLC, as described above. The weight was detected using an analytical balance. The weight loss rate is equal to the weight before incubation minus the weight after incubation and then multiplied by 100%.

### 4.6. Surface Analysis of PET Film

After incubation in LB broth, PET films were washed with 2% SDS, soaked in 70% ethanol for 4 h, washed three times with sterile water, and dried in the oven at 50 °C. The films were weighed using an analytical balance. The weight loss rate is equal to the weight before incubation minus the weight after incubation and then multiplied by 100%. The WCA of the PET film was measured by Kruss DSA100 WCA (Kruss, Hamburg, Germany). To observe the surface changes of the PET film, the films were cut into 5 mm × 5 mm, and the surface was sprayed with gold for SEM observation (Hitachi SU8010 SEM, Hitachi, Japan).

### 4.7. Preparation of Crude Enzyme

Strain YX8 was inoculated into a 50 mL LB broth and grown at 45 °C and 200 rpm for 72 h. The cell-free supernatant of the culture was obtained via centrifugation. The proteins in the supernatant were precipitated using saturated ammonium sulfate. After centrifugation at 10,000× *g* for 30 min, the extracellular enzymes were resuspended in 1 mL of 50 mM Tris-HCl (pH 8.0, 300 mM NaCl). The extracellular enzymes were dropped onto PCL plates at 45 °C for 12 h. A total of 500 μL of the extracellular enzymes was incubated with 15 mg of PET nanoparticles at 45 °C for 48 h and then detected using HPLC. Most of the reported PET degradation enzymes have esterase activity, and an esterase assay is usually performed and used to quantify these enzymes; therefore, the esterase activity in the crude enzymes of YX8 was detected according to previous reports [22,49,50,51,52,53]. The esterase assay system included 10 μL of 10 mM *p*-nitrophenol acetate, 10 μL of crude enzyme, and 980 μL of 50 mM Tris-HCl (pH 8.0 and 300 mM NaCl). The reaction was carried out at 25 °C for 3 min, and then OD_405_ was detected using a spectrophotometer. The activity was calculated according to the previous report [52,53].

## 5. Conclusions

A new method was developed for the isolation of PET-degrading strains. A PET-degrading bacterium *B. safensis* YX8 was obtained from the soil of the surface of PET waste. This is the first reported *B. safensis* strain that can degrade PET. Strain YX8 could hydrolyze PET nanoparticles to generate MHET, BHET, and TPA. After incubation with strain YX8, erosion spots were observed on the PET film, and the WCA of the PET film surface was increased. Moreover, some extracellular enzymes of strain YX8 were involved in the hydrolysis of PET.

## Figures and Tables

**Figure 1 ijms-24-16434-f001:**
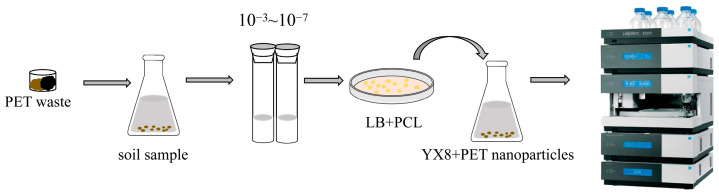
Screening process for PET-degrading strains. The soil on the surface of PET waste was washed, diluted, and spread on the LB plate containing 0.44% PCL. PCL is insoluble in the plate. A transparent halo will arise around its colony when a strain is able to hydrolyze PCL. The PET-degrading ability of this kind of strain was verified using PET nanoparticles as substrates, and the products were detected using HPLC.

**Figure 2 ijms-24-16434-f002:**
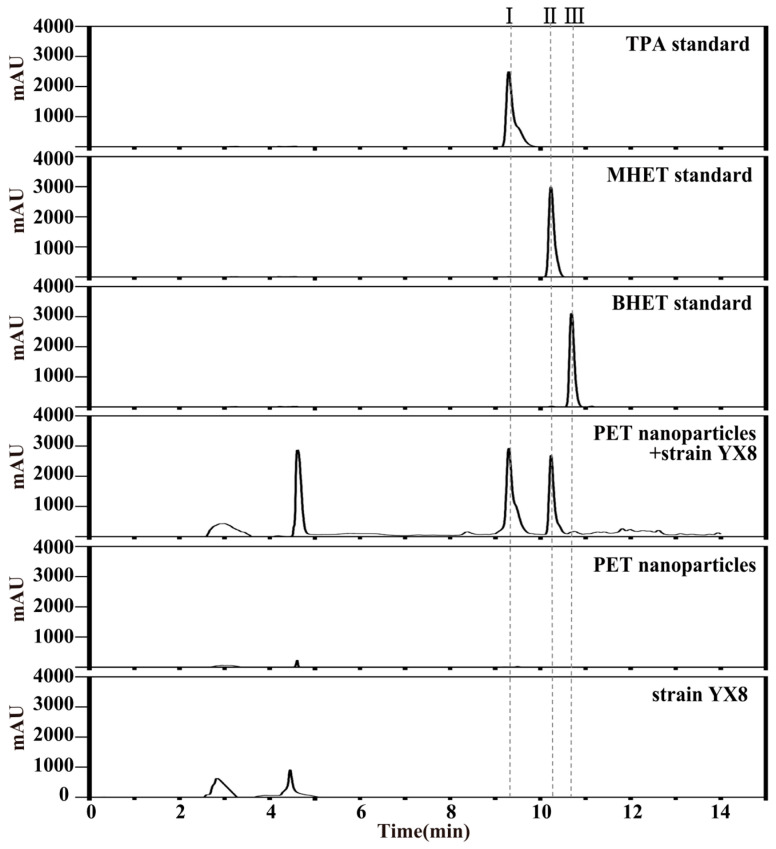
HPLC profile of PET nanoparticles degradation products. PET nanoparticles were incubated with strain YX8 for 15 days at 45 °C, and the products were detected using HPLC. I, II and III represent three products of PET nanoparticles degraded by strain YX8.

**Figure 3 ijms-24-16434-f003:**
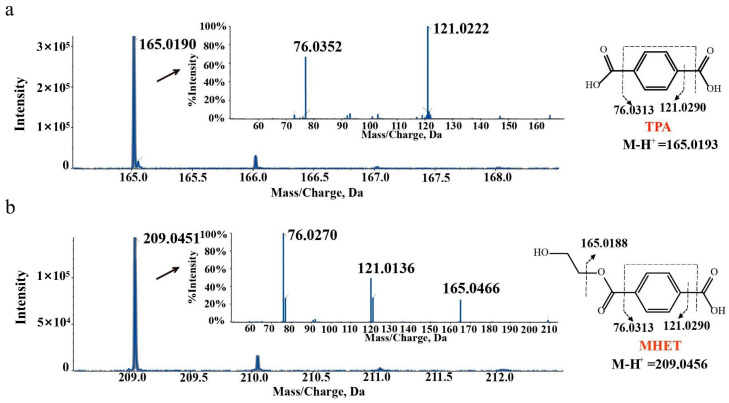
LC-MS/MS profile of PET nanoparticle degradation products. (**a**) LC-MS/MS profile of metabolite A (TPA). (**b**) LC-MS/MS profile of metabolite B (BHET). PET nanoparticles were incubated with strain YX8 for 15 days at 45 °C and detected using LC-MS/MS.

**Figure 4 ijms-24-16434-f004:**
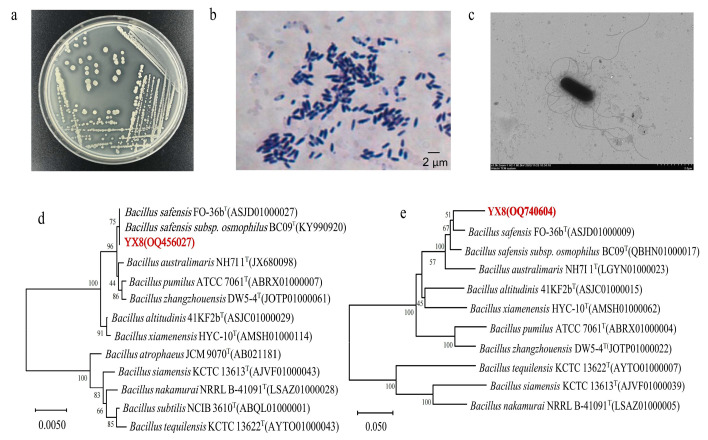
Identification of strain YX8. (**a**) Colonies of strain YX8 grown on a PCL plate after 24 h at 45 °C. (**b**) Gram-staining diagram of strain YX8. (**c**) Transmission electron microscope (TEM) diagram of strain YX8. (**d**) Phylogenetic tree based on 16S rRNA gene sequences of strain YX8. (**e**) Phylogenetic tree based on *gyrA* sequences of strain YX8, obtained using the neighbor-joining method. The numbers at the nodes represent values for bootstrap probabilities.

**Figure 5 ijms-24-16434-f005:**
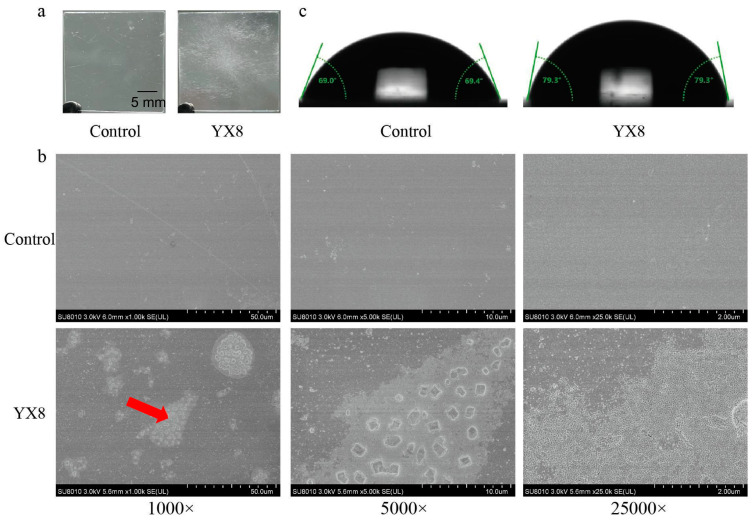
The changes on the surface of PET film after incubation with strain YX8. (**a**) Morphology of PET films (2 cm × 2 cm). (**b**) Scanning electron micrograph of PET films. (**c**) Water contact angle on PET films. PET film was incubated with (YX8) and without (control) strain YX8 in LB broth at 45 °C for 2 months. The film in the control was transparent and smooth and had little change. The red arrow indicates the area to be detected at 5000 times magnification.

**Figure 6 ijms-24-16434-f006:**
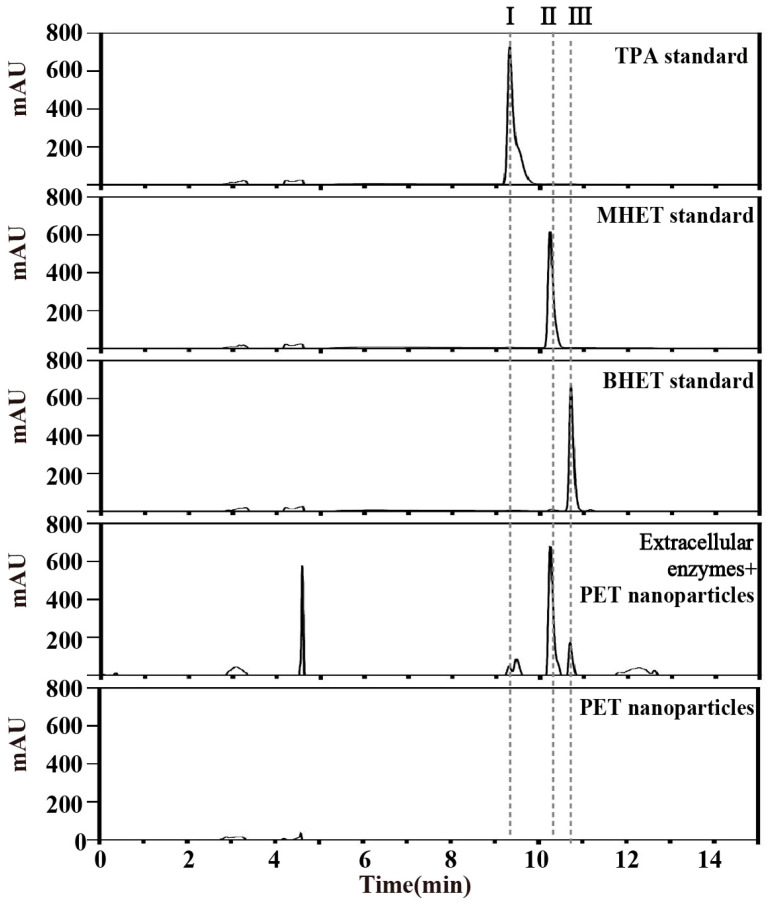
Biodegradation of PET nanoparticles via the extracellular enzymes of strain YX8. The HPLC profiles of PET nanoparticle degradation products after incubation of extracellular enzymes at 45 °C for 48 h. I, II and III represent three products of PET nanoparticles degraded by extracellular enzymes.

## Data Availability

No new data were created or analyzed in this study. Data sharing is not applicable to this article.

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
