# Peer review of "Biodegradation of Poly(ethylene terephthalate) by Bacillus safensis YX8"

_ijms, 2023, doi:10.3390/ijms242216434_

Round 1

Reviewer 1 Report

Comments and Suggestions for Authors

The authors have investigated the use of a new PET degrading strain, YX8, for the rapid, low-temperature depolymerization into less harmful chemical species.  The species was isolated from the surface of PET waste, and the strain tested against PCL and PET nanoparticles as models. HPLC and LC:MS analysis confirmed enzymatic degradation of PET nanoparticles incubated with YX8 after 15 days.

Comments: The approach taken by the authors is consistent with other academic studies that have identified other PET degrading strains.  The methodology is generally sound, and the findings of degradation are not in dispute.  Efforts into identifying the strain were substantial. There are few problems overall.

Grammer:  The technical language is generally acceptable, but the grammar/spelling in several areas could be polished.

Comparative analysis:  In evaluating a new strain, why wasn’t a control strain known to degrade PET also tested so that there could be direct comparative analysis with previous studies and other strains.  At the very least, the authors should have created a table for comparison to other studies and identify the benefits/shortfalls of their approach.

The biodegradation of PET film by strain YX8 was with a commercial film of unknown crystallinity. There are many ways to evaluate this, including manufacturers spec or by DSC methods. This as the authors note, is a problem in achieving more rapid degradation.  As well, comparison with other studies using similar PET films with similar levels of crystallinity was not established. The spotting and erosion on these films is generally unconvincing.

Specific comments:

Introduction:

Line 41 “TfH was purified from Thermobifida fusca, which could break down 50% of PET bottle with 10% crystallinity within 3 weeks….” TfH is a hydrolase produced by Thermobifida fusca abbreviated as TfH and thus stating “TfH was purified from Thermobifida fusca” is redundant, as TfH already indicates that it is an enzyme isolated from Thermobifida fusca. Perhaps state that a “Thermobifida fusca hydrolase (TfH) was purified”.

Lines 45-48 there is no needs to capitalize “metagenomics”, unless it is at the beginning of a sentence

Lines 47-48  no need to capitalize “Leaf Branch Compost”.

Lines 53 “…most highly efficient PET degradation enzymes belong to cutinase” Do the authors mean that most highly efficient PET degradation enzymes are cutinases

Lines 58-59 “Although a variety of PET degradation strains and enzymes have been explored, the high degree of polymerization and non-degradability of PET….” This sentence contains contradictory information. On one hand, the authors state that there are PET-degrading strains and enzymes; on the other hand, the authors state that PET is “non-degradable”.

Lines 60-61 “It is still necessary to explore more novel PET-degrading strains and enzymes to enrich PET-degrading resources” - “more novel” is vague. What kinds of improved characteristics of PET-degrading strains and enzymes are needed?

Materials and methods

Materials and methods lack details.

Line 187 Was MSM used without carbon sources?

Line 198 Where was the PET-contaminated soil collected (what was the source) and how?

Line 199 Why was LB (a nutritionally rich medium with carbon sources) used? What was the anticipated mechanism for PET degradation (e.g. use of PET as a carbon source)?

Line 200 Why was 45°C used for incubation? Was the expectation that bacterial PET degrading tsrains will be thermophilic? If so, why?

Lines 200-201 “The plates were incubated at 45°C to screen the colony that was able to form clear zone around it.” What does “it” refer to?

Lines 201-202 “The strain that showed PCL-degrading capability was inoculated into LB medium containing 15 mg PET nanoparticles…..“ How were strains with PCL-degrading capabilities identified? Why was LB medium used? Why was incubation carried out in LB for 15 days?

Lines 216-217 – please provide a reference for the primers. The authors missed the prime (‘) symbols in “5-AGAGTTTGATCCTGGCTCAG-3 and 5-GGTTACCTTGTTACGACTT-3”. What PCR conditions were used to amplify the 16SrRNA gene?

Lines 218-219: The reference provided for the gyrA gene (“Xu Z, Liu Y, Stefanic P, Miao Y, Xue Y, Xun W, et al. Housekeeping gene gyrA, a potential molecular marker for Bacillus ecology study. AMB Express. 2022;12(1):1-12”) does not reflect the author sequence of the published paper (Liu Y, Štefanič P, Miao Y, Xue Y, Xun W, Zhang N, Shen Q, Zhang R, Xu Z, Mandic-Mulec I.; https://amb-express.springeropen.com/articles/10.1186/s13568-022-01477-9 ). 

Lines 218-219: The authors missed the prime (‘) symbols in gyrA-F “5-GCDGCHGCNATGCGTTAYAC-3” and gyrA-R (5-ACAAGMTCWGCKATTTTTTC-3”. What PCR conditions were used for amplification?

Lines 222 - 227 “Strain YX8 was incubated in LB broth with 0.14 g (2 cm×2 cm) PET film at 45 and 223

200 rpm for 2 months. Fresh LB broth was added weekly.” What was the broth volume and was the volume kept constant (was spent medium removed as new medium was added)? What volume of fresh medium was added weekly? How many PET film squares were added to the incubation vessel? Why was the PET film washed with SDS prior to the addition to the LB broth? Why was LB broth used? Was an abiotic control set up?

Line 236 What was the purpose of the esterase assay?

How were SEM images generated and how were samples prepared for SEM? Please include a section on SEM.

Section 2 Results

Lines 133 “However, no degradation products such as TPA, MHET and BHET were detected by HPLC, and no significant weight loss was observed”. How was weight loss determined? (it is not mentioned in the methods section). The authors state no “significant” weight loss was observed – does that mean that the weight loss of films in controls were compared to the weight loss of films in test setups and the comparison yielded no significant differences? Can the data be show?

Lines 136-139

Figure 5a. How were the images in 5a obtained? There is no apparent film in the left image of Fig. 5a (control). What did the authors wish to illustrate with the image from the control set up?

Figure 5b. There are no apparent films in the control images of Fig. 5b. What did the authors wish to illustrate with the images from the control set up? For the YX8 images, what does the red arrow point to? It is difficult to see “multiple erosion spots on the surface of the film, and there were multiple massive structures on the surface of the erosion spot”, as there is no apparent film in the control images (for comparison purposes). Also, it is not possible to clearly identify erosion spots in the 5000x magnification image for YX8 ( “At 5000 times magnification, it was observed that the erosion spot was clearly demarcated from the surrounding area, and further magnification revealed that the PET film was cracked at these erosion spots”) as there is no film shown in the corresponding control image.

It would be very useful if the authors provided scale bars in the images shown in Fig. 5.

Section 2.4

Lines 143-146 “Esterase enzyme activity assay indicated extracellular esterase activity was highest at 72 h (data not shown), so the extracellular enzymes at this time was concentrated and incubated with PCL plate and PET nanoparticles. The concentrated enzymes exhibited the ability to form clear zone on PCL plate within 12 h (data not shown).” The data can be shown as supplementary figures.

Section 3.0 Discussion

Line 170 “…….and PET nanoparticles were more readily biodegradable than PET film” Could the authors indicate what data supports this statement?

Line 174 “……and cause obvious erosion traces on the surface of PET film,…” Fig. 5 did not show “obvious erosion traces”.

Lines 174 to 176 “…….no significant weight loss was found when using PET film as substrate, indicating that the enzymes involved have low activity toward crystallized PET.” Can the authors speculate what could be possible causes for low enzymatic activity on crystallized PET? Are there any comparable published findings? 

Lines 178-179 “It is speculated that the increase of roughness on the surface led to the rise of hydrophobicity”. Perhaps the authors can cite https://onlinelibrary.wiley.com/doi/10.1111/jicd.12125, which concluded that surface roughness increased hydrophobicity.

Conclusions. Line 243. The description of the isolation approach (section 4.3) does not indicate the PET degrading strain was isolated from “the surface” of PET waste.

Final comments:  I think the manuscript requires a significant rework that ties their work more closely with other previous studies. Otherwise it comes across as a bit of an outlier. The language and word choices and grammar must also be significantly improved. I recommend major revisions.

Comments on the Quality of English Language

It was fair in most places. Still needs to be polished in several places

Author Response

We appreciate your very detailed and constructive suggestions on our manuscript. We have studied comments carefully and have made correction. The corrections and the responses to the comments are as following.

Q1: Comparative analysis: In evaluating a new strain, why wasn’t a control strain known to degrade PET also tested so that there could be direct comparative analysis with previous studies and other strains. At the very least, the authors should have created a table for comparison to other studies and identify the benefits/shortfalls of their approach.

A: Thanks for your suggestions. Yes. It is good to have a positive control. However, the materials and methods employed in this work are commonly used in other reports. Therefore, it is possible to compare our work to other reports. According to your suggestion, a table (in the supplementary materials) was created for comparison to other studies. And the comparison was also added in the discussion section.

Q2: The biodegradation of PET film by strain YX8 was with a commercial film of unknown crystallinity. There are many ways to evaluate this, including manufacturers spec or by DSC methods. This as the authors note, is a problem in achieving more rapid degradation. As well, comparison with other studies using similar PET films with similar levels of crystallinity was not established. The spotting and erosion on these films is generally unconvincing.

A: Thanks for your suggestion. The crystallinity of the commercial PET film has added in the revised manuscript (Lines 223). Comparison with other studies was added in the discussion section. Compared to the control, spotting and erosion on the films are obvious, which is explained in the following response.

Specific comments:

Introduction:

Q1: Lines 41 “TfH was purified from Thermobifida fusca, which could break down 50% of PET bottle with 10% crystallinity within 3 weeks….” TfH is a hydrolase produced by Thermobifida fusca abbreviated as TfH and thus stating “TfH was purified from Thermobifida fusca” is redundant, as TfH already indicates that it is an enzyme isolated from Thermobifida fusca. Perhaps state that a “Thermobifida fusca hydrolase (TfH) was purified”.

A: Thanks for your suggestion. This description was modified according to your advice. (Lines 42)

Q2: Lines 45-48 there is no needs to capitalize “metagenomics”, unless it is at the beginning of a sentence

A: Thank you very much for your suggestion. This word was modified. (Lines 46)

Q3: Lines 47-48 no need to capitalize “Leaf Branch Compost”.

A: Thank you very much for your suggestion. This description was modified. (Lines 49)

Q4: Lines 53 “…most highly efficient PET degradation enzymes belong to cutinase” Do the authors mean that most highly efficient PET degradation enzymes are cutinases

A: Thanks. This description is ambiguous and not correct. Therefore, we deleted this sentence in the new version.

Q5: Lines 58-59 “Although a variety of PET degradation strains and enzymes have been explored, the high degree of polymerization and non-degradability of PET….” This sentence contains contradictory information. On one hand, the authors state that there are PET-degrading strains and enzymes; on the other hand, the authors state that PET is “non-degradable”.

A: Good question. “non-degradability” was modified to “resistance to biodegradation”. (Lines 57)

Q6: Lines 60-61 “It is still necessary to explore more novel PET-degrading strains and enzymes to enrich PET-degrading resources” - “more novel” is vague. What kinds of improved characteristics of PET-degrading strains and enzymes are needed?

A: Good question. The description “The enzymes and strains that exhibit outstanding ability to degrade crystallized PET or to work efficiently at moderate temperature are desired” was added in the new version. (Lines 60-61)

Materials and methods

Q1: Line 187 Was MSM used without carbon sources?

A: Yes. The MSM medium contains no carbon source, which was used to wash out and dilute the soil sample. MSM medium contains NH4NO3 1.0 g/L,KH2PO4 0.5 g/L,K2HPO4 1.5 g/L,NaCl 0.5 g/L,MgSO4·7H2O 0.2 g/L. (Lines 226)

Q2: Line 198 Where was the PET-contaminated soil collected (what was the source) and how?

A: PET waste was collected from a landfill plant in Xuancheng City, Anhui Province, China and the soil on the surface of PET waste was used to isolate PET-degrading bacteria. This information is shown in lines 239.

Q3: Line 199 Why was LB (a nutritionally rich medium with carbon sources) used? What was the anticipated mechanism for PET degradation (e.g. use of PET as a carbon source)?

A: Good question. So far, only one PET-degrading microbe (Ideonella sakaiensis 201-F6) could grow on PET, while others just depolymerize PET and fail to grow on the intermediates. In the natural environment, it is likely that PET and its intermediates are degraded via the cooperation of multiple microbes; some strains just depolymerize PET and others utilize the intermediates. It is difficult to obtain the strain that can grow on PET. Therefore, we used the a nutritionally rich medium to isolate the strains that are able to depolymerize PET.

Q4: Line 200 Why was 45°C used for incubation? Was the expectation that bacterial PET degrading strains will be thermophilic? If so, why?

A: Good question. We expect to obtain the PET-degrading enzyme that exhibit high activity at moderate temperature such as 45°C. Now most efficient enzymatic depolymerization of PET is performed at high temperature (about 70°C), which will consume much energy in the industrial recycling of PET. The enzymes that enable highly efficient depolymerization of PET at moderate temperature are desired for the industrial recycling of PET.

Q5: Lines 200-201 “The plates were incubated at 45°C to screen the colony that was able to form clear zone around it.” What does “it” refer to?

A: Thank you very much for your suggestion. “it” refers to the colony. This sentence was modified to “the strain that was able to form transparent zone around its colony.” in lines 243.

Q6: Lines 201-202 “The strain that showed PCL-degrading capability was inoculated into LB medium containing 15 mg PET nanoparticles…..“ How were strains with PCL-degrading capabilities identified? Why was LB medium used? Why was incubation carried out in LB for 15 days?

A: Good question.

(1) If a strain can hydrolyze PCL, a transparent halo will be formed around its colony. This method has been verified and used in previous reports (references [30]).

(2) The strain was unable to grow on PET, and LB was added to support the growth of the strain.

(3) The degradation capacity of the strains is generally not strong, so the incubation time was extended.

Q7: Lines 216-217 – please provide a reference for the primers. The authors missed the prime (‘) symbols in “5-AGAGTTTGATCCTGGCTCAG-3 and 5-GGTTACCTTGTTACGACTT-3”. What PCR conditions were used to amplify the 16SrRNA gene?

A: Thanks for your suggestion. The reference and the PCR conditions was provided in the revised version. (Lines 273-277) And the (‘) has be added.

Q8: Lines 218-219: The reference provided for the gyrA gene (“Xu Z, Liu Y, Stefanic P, Miao Y, Xue Y, Xun W, et al. Housekeeping gene gyrA, a potential molecular marker for Bacillus ecology study. AMB Express. 2022;12(1):1-12”) does not reflect the author sequence of the published paper (Liu Y, Štefanič P, Miao Y, Xue Y, Xun W, Zhang N, Shen Q, Zhang R, Xu Z, Mandic-Mulec I.;https://amb-express.springeropen.com/articles/10.1186/s13568-022-01477-9).

A: Thanks. This mistake was modified.

Q9: Lines 218-219: The authors missed the prime (‘) symbols in gyrA-F “5-GCDGCHGCNATGCGTTAYAC-3” and gyrA-R (5-ACAAGMTCWGCKATTTTTTC-3”. What PCR conditions were used for amplification?

A: Thanks for your suggestion. The reference and the PCR conditions was provided in the revised version. (Lines 273-277) And the (‘) has be added.

Q10: Lines 222 - 227 “Strain YX8 was incubated in LB broth with 0.14 g (2 cm×2 cm) PET film at 45 ℃and 200 rpm for 2 months. Fresh LB broth was added weekly.” What was the broth volume and was the volume kept constant (was spent medium removed as new medium was added)? What volume of fresh medium was added weekly? How many PET film squares were added to the incubation vessel? Why was the PET film washed with SDS prior to the addition to the LB broth? Why was LB broth used? Was an abiotic control set up?

A: Good questions.

(1) The broth volume was 50 ml. (Lines 284) The volume was kept constant via adding fresh LB broth weekly.

(2) One PET film square was added in each flask and three repeats were set up. (Lines 285)

(3) In order to clear the PET film and keep sterile, PET films were washed with 2% SDS, followed by soaking in 70% ethanol for 4 h. (Lines 283)

(4) The strain was unable to grow on PET, and LB was added to support the growth of the strain.

(5) Yes. An abiotic control was set up with three repeats. (Lines 286)

Q11: Line 236 What was the purpose of the esterase assay?

A: Good question. Most of reported PET-degrading enzymes have esterase activity [22, 49-53]. It is easy to quantified the enzyme using the esterase assay. (Lines 305-308)

Q12: How were SEM images generated and how were samples prepared for SEM? Please include a section on SEM.

A: Thanks for your suggestion. The sample prepared method has mentioned in the section 4.6 (Lines 297-299)

Section 2 Results

Q1: Lines 133 “However, no degradation products such as TPA, MHET and BHET were detected by HPLC, and no significant weight loss was observed”. How was weight loss determined? (it is not mentioned in the methods section). The authors state no “significant” weight loss was observed – does that mean that the weight loss of films in controls were compared to the weight loss of films in test setups and the comparison yielded no significant differences? Can the data be show?

A: Good questions. The information was added in the methods section (Lines 287-290) and results section (Lines 146-149). The weight was detected by analytical balance. The weight loss rate is equal to the weight before incubation minus the weight after incubation and then multiplied by 100%. Yes. The weight loss of films in controls were compared to the weight loss of films in test setups and the comparison yielded no significant differences. The weight loss of films in test setups was slightly higher than that in controls, but not significant, which was shown in the supplementary materials (Figure. S1).

Q2: Lines 136-139 Figure 5a. How were the images in 5a obtained? There is no apparent film in the left image of Fig. 5a (control). What did the authors wish to illustrate with the image from the control set up?

A: Actually, there is a film in the control; since the film is transparent and smooth, it looks like there is no film. In contrast, there were many scratches on the surface of the PET film treated with strain YX8.

Q3: Figure 5b. There are no apparent films in the control images of Fig. 5b. What did the authors wish to illustrate with the images from the control set up? For the YX8 images, what does the red arrow point to? It is difficult to see “multiple erosion spots on the surface of the film, and there were multiple massive structures on the surface of the erosion spot”, as there is no apparent film in the control images (for comparison purposes). Also, it is not possible to clearly identify erosion spots in the 5000x magnification image for YX8 (“At 5000 times magnification, it was observed that the erosion spot was clearly demarcated from the surrounding area, and further magnification revealed that the PET film was cracked at these erosion spots”) as there is no film shown in the corresponding control image.

A: Thanks for your suggestion. Actually, there is a PET film in the control; since the film is transparent and smooth, it looks like there is no film. Compared to the film in the control, erosion spots were obvious in the treatment with strain YX8. Red arrow points to the field of view at 5000×.

Q4: It would be very useful if the authors provided scale bars in the images shown in Fig. 5.

A: Thanks for your suggestion. The scale bars were added.

Section 2.4

Q1: Lines 143-146 “Esterase enzyme activity assay indicated extracellular esterase activity was highest at 72 h (data not shown), so the extracellular enzymes at this time was concentrated and incubated with PCL plate and PET nanoparticles. The concentrated enzymes exhibited the ability to form clear zone on PCL plate within 12 h (data not shown).” The data can be shown as supplementary figures.

A: Thanks for your good suggestion. We have supplement figure S2 and S3 in the supplementary materials (Figure S2 S3).

Section 3.0 Discussion

Q1: Line 170 “…….and PET nanoparticles were more readily biodegradable than PET film” Could the authors indicate what data supports this statement?

A: Good question. PET nanoparticles have low crystallinity and high specific surface area, which are more readily biodegradable than PET film. This phenomenon was verified by previous reports (references 35), which were added in the revised version. (Lines 202-203)

Q2: Line 174 “……and cause obvious erosion traces on the surface of PET film,…” Fig. 5 did not show “obvious erosion traces”.

A: Actually, there is a film in the control; since the film is transparent and smooth, it looks like there is no film. Compared to the film in the control, erosion spots were obvious in the treatment with strain YX8.

Q3: Lines 174 to 176 “…….no significant weight loss was found when using PET film as substrate, indicating that the enzymes involved have low activity toward crystallized PET.” Can the authors speculate what could be possible causes for low enzymatic activity on crystallized PET? Are there any comparable published findings?

A: Good question. One possibility is that the enzymes responsible for PET depolymerization in strain YX8 have low activity toward crystallized PET. Alternative reason is that the expression level of these enzymes is very low. This issue was mentioned in the discussion section.

Q4: Lines 178-179 “It is speculated that the increase of roughness on the surface led to the rise of hydrophobicity”. Perhaps the authors can cite https://onlinelibrary.wiley.com/doi/10.1111/jicd.12125, which concluded that surface roughness increased hydrophobicity.

A: Thanks. This reference was cited in the revised manuscript. (Lines 218)

Conclusions.

Q1: Line 243. The description of the isolation approach (section 4.3) does not indicate the PET degrading strain was isolated from “the surface” of PET waste.

A: Thank you very much for your suggestion. This content has been modified in the method. (Lines 240)

Q: Final comments: I think the manuscript requires a significant rework that ties their work more closely with other previous studies. Otherwises it comes across as a bit of an outlier. The language and word choices and grammar must also be significantly improved. I recommend major revisions.

A: Again, we appreciate your very detailed and constructive suggestions on our manuscript. The manuscript is significantly improved after the modification according to your suggestion. The language was also revised by an expert who are good at scientific article writing.

Reviewer 2 Report

Comments and Suggestions for Authors

The manuscript entitled Biodegradation of poly (ethylene terephthalate) by Bacillus safensis YX8 is very interesting for the scientific community.  However, some revisions should be made by the authors to the manuscript. My comments to the authors are listed below.

Abstract: Please replace “microbes” with another proper word. 

Materials and methods:  Please mention how many times you repeated the experiments.

The authors mention that “The morphology of strain YX8 was observed by Gram staining and SEM.” Please describe briefly how the samples were prepared for SEM analysis. What SEM equipment was used?

Please add more details about PCR amplification and sequencing of 16S rRNA and gyrA genes. What types of equipment did you use for PCR and sequencing? Add also the accession number for the isolated bacteria.

Results and Discussion:  Please replace “microbes” with another word. 

Please add the scale bar on the microscopy images (Figure 4).

Although the authors say in the material and methods and also in the title of Figure 4  that they used SEM, picture 4c is a TEM image.

In Figure 5, the SEM images are not very clear. Please add more pictures at higher magnification in order to see the PET surface changes as compared with the control.

Comments on the Quality of English Language

-

Author Response

We appreciate your very detailed and constructive suggestions on our manuscript. We have studied comments carefully and have made correction. The corrections and the responses to the comments are as following.

Q1: Abstract: Please replace “microbes” with another proper word.

A: Thank you very much for your suggestion. The word “microbes” has been replaced to “strains”. (Lines 15)

Q2: Materials and methods: Please mention how many times you repeated the experiments.

A: Thank you very much for your suggestion. All experiments are set up in triplicate and have mentioned in the materials and methods.

Q3: The authors mention that “The morphology of strain YX8 was observed by Gram staining and SEM.” Please describe briefly how the samples were prepared for SEM analysis. What SEM equipment was used?

A: Thank you very much for your suggestion. How the samples were prepared and what SEM equipment was used has been supplied in the manuscript. (Lines 297-299)

Q4: Please add more details about PCR amplification and sequencing of 16S rRNA and gyrA genes. What types of equipment did you use for PCR and sequencing? Add also the accession number for the isolated bacteria.

A: Thank you very much for your suggestion. The details of PCR amplification and sequencing of 16S rRNA and gyrA genes has been added. (Lines 275-277) The PCR equipment type is T20 multi-block thermal cycler. The accession number of strain YX8 has added in the phylogenetic trees and the sequences of 16S rRNA and gyrA genes were provided in the supplementary materials.

Q5: Results and Discussion: Please replace “microbes” with another word.

A: Thank you very much for your suggestion. The word “microbes” has been replaced to “strains”.

Q6: Please add the scale bar on the microscopy images (Figure 4).

Thank you very much for your suggestion. The scale bar has added on the microscopy images.

Q7: Although the authors say in the material and methods and also in the title of Figure 4 that they used SEM, picture 4c is a TEM image.

A: Thank you very much for your suggestion. This mistake was modified.

Q8: In Figure 5, the SEM images are not very clear. Please add more pictures at higher magnification in order to see the PET surface changes as compared with the control.

A: Thank you very much for your suggestion. The pictures already had high resolution rate. Notably, there is a film in the control; since the film is transparent and smooth, it looks like there is no film. In contrast, there were many scratches on the surface of the PET film treated with bacteria.

Thank you again for your valuable comments on this draft. I have revised the paper in detail according to the comments.

Round 2

Reviewer 1 Report

Comments and Suggestions for Authors

No new comments. The authors have sufficiently answered my questions

Comments on the Quality of English Language

Sufficient

Reviewer 2 Report

Comments and Suggestions for Authors

The manuscript entitled Biodegradation of poly (ethylene terephthalate) by Bacillus safensis YX8” has been improved according to the referees' instructions and could be accepted for publication.